# Combined Hepatocellular-Cholangiocarcinoma: An Update on Pathology and Diagnostic Approach

**DOI:** 10.3390/biomedicines10081826

**Published:** 2022-07-29

**Authors:** Joon Hyuk Choi, Jae Y. Ro

**Affiliations:** 1Department of Pathology, College of Medicine, Yeungnam University, Daegu 42415, Korea; 2Department of Pathology and Genomic Medicine, Houston Methodist Hospital, Weill Medical College of Cornell University, Houston, TX 77030, USA; jaero@houstonmethodist.org

**Keywords:** combined hepatocellular-cholangiocarcinoma, hepatocellular carcinoma, cholangiocarcinoma, liver cancers, adult

## Abstract

Combined hepatocellular-cholangiocarcinoma (cHCC-CCA) is a rare primary liver carcinoma displaying both hepatocytic and cholangiocytic differentiation within the same tumor. Relative to classic hepatocellular carcinoma (HCC), cHCC-CCA has more aggressive behavior and a poorer prognosis. Though recent advances have improved our understanding of the biology underlying cHCC-CCAs, they remain diagnostically challenging for pathologists because of their morphologic and phenotypic diversity. Accurate diagnosis of cHCC-CCA is important for patient management and prognostication. Herein, we review recent updates on cHCC-CCA, focusing on tumor classification, pathology, and diagnostic approach.

## 1. Introduction

Combined hepatocellular-cholangiocarcinoma (or combined hepatocellular carcinoma and cholangiocarcinoma; cHCC-CCA) is a primary liver carcinoma (PLC) defined by the unequivocal presence of both hepatocytic and cholangiocytic differentiation [1]. cHCC-CCA accounts for 2–5% of PLCs [2] and is also referred to as mixed HCC-CCA, mixed hepatobiliary carcinoma, hepatocholangiocarcinoma, and biphenotypic PLC. The age of onset, sex specificity, clinical features, and geographic distribution are similar to conventional HCC and intrahepatic CCA (iCCA) [1].

The definition and related terminology of cHCC-CCA have evolved since the tumor was first described by Wells in 1903 [3]. Improvements in ancillary techniques to identify hepatocytic and cholangiocytic differentiation have driven clinical recognition of a spectrum of PLCs with mixed and variable differentiation. Brunt et al. [2] proposed a consensus terminology for PLCs displaying both hepatocytic and cholangiocytic differentiation to unify histological approaches for diagnostic and research purposes and facilitate scientific studies. In the 5th edition of the 2019 World Health Organization (WHO) of digestive system tumors, cHCC-CCA is more clearly defined and distinguished from other entities [4]. However, cHCC-CCA remains challenging to diagnose and treat because of its complex morphological and phenotypic diversity [5,6,7]. Herein, we review recent updates on cHCC-CCA, focusing on tumor pathology, immunohistochemical features, diagnostic approaches, and differential diagnosis.

## 2. Clinical Features

Clinically, cHCC-CCA most commonly presents as a mass lesion in the liver with possible abdominal discomfort, fatigue, weight loss, abdominal pain, and obstructive jaundice [8]. Patients may also be asymptomatic. A review of 465 patients with cHCC-CCA analyzed from 1988 to 2009 using the Surveillance, Epidemiology, and End Results database indicated that the tumor most commonly arises in white, male patients over 65 years of age [9]. In approximately 15% of patients with cHCC-CCA, alpha-fetoprotein (AFP) and carbohydrate antigen 19-9 (CA 19-9) are both elevated [10].

## 3. Radiological Features

Radiologic features of cHCC-CCA reflect the predominant histological features of HCC and CCA in the same nodule [11,12]. Tumors with predominant HCC components may show signal enhancement in the arterial phase (hyperintensity) and signal dropout (hypointensity) in the delayed venous phase on contrast-enhanced CT and MRI, similar to HCCs (Figure 1). In contrast, lesions with predominant CCA components demonstrate peripheral rim enhancement in the arterial phase with progressive centripetal enhancement in the delayed venous phase, similar to mass-forming CCAs [13,14,15,16]. Potretzke et al. [17] demonstrated that most cHCC-CCAs had features of non-HCC malignancy and that the addition of ancillary features favoring non-HCC malignancy to major features of HCC may improve diagnostic accuracy over systems in which only major features are used.

Wells et al. [18] demonstrated that biphenotypic hepatic tumors can be suggested when imaging findings or tumor markers (AFP and CA19-9) are discordant with the most likely diagnosis based on enhancement pattern. Features including satellite nodules, hyperintense signal on T_2_-weighted images, restricted diffusion, and the absence of capsule appearance on MRI are suggestive of cHCC-CCA [19]. The Liver Imaging Reporting and Data System (LI-RADS) is a comprehensive and dynamic system for interpreting and reporting hepatic observations in patients at high risk for HCC [20]. Zou et al. [21] showed that only 41.67% cHCC-CCAs were assigned as LR-M (LR-M; probably or definitely malignant but not specific for HCC), and nearly half of cHCC-CCAs were categorized as LR-5 (LR-5; definitely HCC). Zhou et al. [22] presented new criteria for diagnosis of cHCC-CCAs by combining contrast-enhancing ultrasound (CEUS) and CT/MRI LI-RADS in association with serum biomarkers. Information on characteristic radiologic features of cHCC-CCAs is limited and future imaging-based studies for cHCC-CCA are warranted.

## 4. Etiology

The specific etiologies associated with the development of cHCC-CCA are unknown; cHCC-CCAs have the same risk factors as HCC, including hepatitis B, hepatitis C infection, and cirrhosis [11,23]. Additionally, a higher frequency of cHCC-CCA is observed in patients following transarterial chemoembolization (TACE) for HCC [24,25]. The prevalence of cHCC-CCA in patients with non-alcoholic fatty liver disease and alcoholic liver disease is not well defined. Further study on the specific etiologies of cHCC-CCA is warranted.

## 5. Pathogenesis

The pathogenesis of cHCC-CCA is not well defined, but three possible pathogenetic processes have been postulated [26]. First, HCC and CCA may arise independently and separately. Second, cHCC-CCAs may originate from stem/progenitor cells that differentiate with both hepatocytes and cholangiocyte lines. Once the malignant transformation of hepatic stem/progenitor cells occurs, cells may differentiate completely or incompletely to HCC and CCA. Theise et al. [27] reported four cases of hepatic “stem cell” malignancies in adults and suggested that these tumors are of hepatic stem/progenitor cell origin. Third, HCC may arise first and transforms into CCA at varying degrees. In this case, the morphologic and phenotypic diversity of cHCC-CCAs would be due to the various differentiation stages of hepatic stem/progenitor cells with malignant transformation to HCC and CCA. Indeed, Li et al. [28] demonstrated that CCA-like traits are acquired during HCC progression in mice models.

## 6. Pathological Features

### 6.1. Historical Background in the Classification of cHCC-CCA

The definitions and terms describing cHCC-CCA have evolved since its first description by Wells in 1903 [3]. In 1949, Allen and Lisa [29] illustrated the dual nature of these tumors as three combinations of (1) separate nodules of hepatocellular and bile duct carcinoma, (2) contiguity with intermingling, and (3) intimate association due to origin from the same focus. In 1954, Edmondson et al. [30] called these tumors “hepatobiliary carcinoma”. In 1985, Goodman et al. [31] classified cHCC-CCA into three subtypes: (1) type I (collision tumors), (2) type II (transitional tumors), and (3) type III (fibrolamellar tumors).

In the 2000 WHO classification (3rd edition), cHCC-CCA was defined as a rare tumor containing unequivocal and intimately admixed elements of both HCC and CCA [32]. In an updated 2010 WHO classification, cHCC-CCAs were divided into a classical type and three subtypes with stem cell features of (1) typical, (2) intermediate cell, and (3) cholangiolocellular subtypes based on their morphological or phenotypical features of stem/progenitor cells [33]. It was uncertain whether there were biological differences between them at that time.

Some studies have shown that stem/progenitor cell features and desmoplastic stroma may be present in many PLCs; thus, these features are no longer considered characteristic of unique diagnostic subtypes of cHCC-CCA [34,35]. In 2018, an international group of hepatic pathologists, radiologists, surgeons, and clinicians proposed a consensus terminology for PLCs with both hepatocytic and cholangiocytic differentiation [2]. They recommended that diagnosis be based on routine histopathology with hematoxylin and eosin (H&E) staining and indicated that immunohistochemical stains are supportive but not essential for diagnosis. Stem/progenitor cell features or phenotypes may exist within cHCC-CCA and can be described in the pathologic report. The key elements of the 2018 consensus are now included in the 2019 WHO classification [1]. Two other entities of PLCs, intermediate cell carcinoma (ICC) and cholangiolocarcinoma (CLC), are also newly defined. Both ICC and CLC may coexist with HCC, CCA, or cHCC-CCA. The evolution of the WHO classification of cHCC-CCA is summarized in Table 1.

### 6.2. Macroscopic Appearance

The macroscopic appearance of cHCC-CCAs is similar to HCCs and, in part, similar to CCAs [36]. The prevalence of major tumor components influences macroscopic appearance. Grossly, the cut surface shows distinctly heterogeneous areas within the mass. In tumors with a major HCC component, the cut surface resembles HCC and is a yellowish, green to tan color. It is usually well-demarcated, soft, and bulges out from the cut surface of the liver, with varying degrees of hemorrhage and necrosis [37]. In tumors with a major CCA component, the cut surface is grayish-white, fibrotic, and firm with an infiltrative tumor border [38].

### 6.3. Histopathology

#### 6.3.1. Combined Hepatocellular-Cholangiocarcinoma

Histologically, cHCC-CCA shows unequivocal features of both HCC and CCA [1]. The typical HCC component shows polygonal tumor cells that resemble hepatocytes, are arranged in a trabecular pattern, and are separated by sinusoids (Figure 2A–C). Bile can be identified. The typical CCA component shows neoplastic glands and abundant desmoplastic stroma. Mucin is often present. Two components are either close to each other or extensively intermingled. There is no consensus regarding a strict cutoff for each component to diagnose cHCC-CCA [1]. The transitional area between the two components often exhibits mixed features with intermediate morphology [33]. Biphenotypic differentiation should be based on H&E morphology. Mucin stains can help confirm glandular differentiation. Immunohistochemistry without supportive morphological features is insufficient for a diagnosis of cHCC-CCA. Distant metastases from cHCC-CCA can show both HCC and CCA components, an HCC component alone, or a CCA component alone [39].

Histologically, stem/progenitor cells show small uniform tumor cells with hyperchromatic nuclei, a high nuclear/cytoplasmic ratio, inconspicuous nucleoli, and scant cytoplasm [1]. Mitotic activity is uncommon. These cells are often found in the transitional zone between the HCC and CCA components, at the periphery of the HCC trabeculae, or as small nests without a specific location. The term “cHCC-CCA with stem cell features” is no longer recommended. Histological features of HCC, CCA, cHCC-CCA, and stem/progenitor cells are summarized in Table 2.

#### 6.3.2. Intermediate Cell Carcinoma

PLCs of the intermediate phenotype (hepatocytes and cholangiocytes) were first described by Robrechts et al. [40] in 1998 and have been continuously reported since [34,35,41]. In the 2010 WHO classification [33], the term ‘cHCC-CCA with stem cell features, intermediate-cell subtype’ was applied to PLCs of intermediate cell phenotype. In 2018, an international group (Brunt et al.) proposed the consensus terminology ‘intermediate cell carcinoma (ICC)’ with a monomorphic tumor consisting of tumor cells smaller than normal hepatocytes (but larger than the stem/progenitor cell phenotype) with intermediate features between hepatocytes and cholangiocytes [2]. The 2019 WHO classification used the terminology ‘intermediate cell carcinoma (ICC)’ [1].

Histologically, ICCs show monotonous morphological features that are intermediate between hepatocytes and cholangiocytes (Figure 3A–C) [1]. At lower magnification, the tumor is homogeneous. At higher magnification, the tumor cells are small, with scant cytoplasm and arranged in cords, strands, trabeculae, and occasional gland-like structures within an abundant fibrous stroma. Dual expression of hepatocytic and cholangiocytic markers in the tumor cells supports the intermediate hepatobiliary phenotype. An ICC diagnosis should be reserved for PLCs in which intermediate features are present en bloc [1]. Focal intermediate (hepatobiliary) tumor cells in cHCC-CCA do not support a diagnosis of ICC. Currently, no consensus on whether ICC is a distinct entity or a histological pattern of cHCC-CCA can be reached, as the data on its molecular and clinical characteristics is limited [1].

#### 6.3.3. Cholangiolocarcinoma

Cholangiolocarcinoma (CLC) was first described by Steiner and Higginson [42] in 1959 and is characterized by relatively monotonous small tumor cells that resemble canals of Hering or cholangioles in a fibrous stroma. In the 2010 WHO classification [33], CLC was described under the category of cHCC-CCA with stem cell features, cholangiolocellular subtype. Though CLC may comprise the entire PLC or vary as a component of an HCC, iCCA, or cHCC-CCA, the term CLC should be applied only to a tumor with over 80% ductular configuration [2]. Immunostain results, mutational profile, and copy number alterations indicate that CLC is similar to iCCA [43]. In the 2019 WHO classification, CLC is regarded as a small-duct subtype of iCCA [44].

Histologically, CLC consists of tumor cells arranged in a tubular, cord-like, anastomosing pattern (antler-like pattern) within a dense, hyalinized stroma (Figure 4A–C) [1]. CLC mimics a ductular reaction-like pattern and may display a replacing growth pattern at its interface with surrounding liver parenchyma. CLC shows no mucin production. Immunostains for cholangiocytic markers, including cytokeratin 7 (CK7) and cytokeratin 19 (CK 19), are positive. Luminal and cytoplasmic expression of epithelial membrane antigen is present in 80% of tumors [4]. CD56 (NCAM) is positive in 40% of tumors.

## 7. Immunohistochemical Features

Immunohistochemistry (IHC) can support PLC classification into cHCC-CCA, ICC, or CLC [2]. In general, IHC can help identify the line of differentiation. Immunohistochemical markers for hepatocytic and cholangiocytic differentiation and stem/progenitor cells are summarized in Table 3. Markers for hepatocytic differentiation include hepatocyte in paraffin 1 (HepPar-1), arginase-1, AFP, glypican-3, polyclonal carcinoembryonic antigen (pCEA), and CD10 [45,46,47,48]. Arginase-1 tends to perform better than HepPar−1 in poorly differentiated HCCs. pCEA and CD10 are the most specific IHC markers when canalicular staining is observed, but their sensitivity for HCC is only ~60–80%. AFP staining is positive in approximately one-third of HCCs. Markers for cholangiocytic differentiation include CK7, CK19, and EpCAM. Approximately 25–30% of HCCs (in particular, those with atypical features such as fibrous stroma, more infiltrative growth, and lymph node metastasis) show positive immunostaining for the cholangiocytic markers CK7 and CK 19 (Figure 5A–C) [49,50].

Stem/progenitor cell immunohistochemical markers include CK19, EpCAM, CD56 (NCAM), CD117 (KIT), and CD133. However, interpretation of these immunostains can be challenging because PLC immunophenotypes overlap with cholangiocytic markers [2]. Many stem/progenitor cell markers, such as CK19, EpCAM, and CD56 (NCAM), are present in cholangiocytes at various stages of development. Thus, the presence of these markers should be interpreted based primarily on morphological characteristics of positive cells [48]. Cells with the morphological characteristics of cholangiocytes forming ductules should be regarded as cholangiocytes. Reactive cells with cellular morphology resembling stem/progenitor cells should be considered stem/progenitor cells. A summary of PLC classifications is provided in Table 4.

Nestin expression is significantly increased in cHCC-CCAs [51], underscoring its potential as a diagnostic and prognostic biomarker for HCC-CCA. Nestin is not expressed by hepatocellular or cholangiocellular components but is expressed by most of the intermediate cells in cHCC-CCA [52]. Sasaki et al. [53] demonstrated that Nestin-positive cHCC-CCAs are characterized by smaller tumor size, more abundant CLC components, higher rate of p53 overexpression, and higher rates of multiple genetic alterations. Thus, nestin may be a useful diagnostic marker for a specific subgroup of cHCC-CCAs and small duct type iCCA associated with CLC components [53].

## 8. Molecular Features

The molecular biology of cHCC-CCA remains poorly characterized because of its rare occurrence and diagnostic complexity. Coulouarn et al. [54] reported that cHCC-CCA exhibited stem/progenitor features, a downregulated hepatocyte differentiation program, and commitment to the biliary lineage. TGF-β and Wnt/β-catenin are two major signaling pathways activated in cHCC-CCA. A study by Fujimoto et al. [55] revealed the strong impact of chronic hepatitis on the mutational landscape in liver cancers and the genetic diversity among liver cancers displaying a biliary phenotype (LCBs). Specifically, the frequencies of *KRAS* and *IDHs* mutations, which are associated with poor disease-free survival, are significantly higher in hepatitis-negative LCBs. Moeini et al. [56] reported that in classical type cHCC-CCA, the copy number variations of HCC and CCA components are significantly correlated, suggesting a clonal origin. In the ‘stem cell features’ subtype of cHCC-CCA, tumors are characterized by spalt-like transcription factor 4 (SALL4) positivity, enrichment of progenitor-like gene signatures, activation of specific oncogenic signaling pathways (i.e., MYC and insulin-like growth factor), and signatures associated with poor clinical outcome.

In genomic and transcriptomic analyses of 133 cHCC-CCA cases, combined type cHCC-CCAs showed strong iCCA-like features, including higher expression of *EpCAM, CK19,* and *PRDM5*, and enrichment of *KRAS* mutations and higher expression of *KRAS* [51]. Joseph et al. [57] reported that cHCC-CCA molecular profiles are similar to HCC, even in the CCA component. cHCC-CCA harbored recurrent alterations in *TERT* (80%), *TP53* (80%), cell cycle genes (40%; *CCND1*, *CCNE1*, *CDKN2A*), RTK/Ras/PI3K/AKT signaling pathway genes (55%; *ERBB2*, *KRAS*, *MET*, *PTEN*), chromatin regulators (20%; *ARID1A*, *ARID2*), and Wnt pathway genes (20%; *APC*, *AXIN*, *CTNNB1*). *TERT* promoter mutations were consistently identified in both HCC and CCA components, supporting *TERT* alteration as an early event in cHCC-CCA evolution. Malvi et al. [52] showed that intermediate areas and HCC areas of cHCC-CCA shared the same mutational profiles and that both harbored different mutations relative to CCA areas. Common recurrent mutations in HCC, CCA, and cHCC-CCA are summarized in Table 5. The molecular profiles and pathological features of cHCC-CCA vary [58]. Future studies are needed to obtain more detailed molecular pathogenetic evidence in cHCC-CCA. The classification of cHCC-CCA may be redefined with new molecular data.

## 9. Diagnostic Approach and Differential Diagnosis

### 9.1. Specimen Handling

#### 9.1.1. Biopsy Specimen

Histopathological diagnosis of cHCC-CCAs is relatively difficult in small biopsy specimens but can be aided by careful imaging evaluation [59]. Radiologically heterogeneous tumors should have their different components biopsied [11]. Needle-core biopsy specimens should be measured and submitted in the entirety for routine histology unless special studies are indicated [59]. In general, step sections are preferred to serial sections so that intervening sections are available for histochemical and immunohistochemical stains.

#### 9.1.2. Partial Hepatectomy Specimen

There are no definite guidelines for pathological assessment of hepatectomy specimens. The liver should be serially sectioned perpendicular to the resection margin using thin (e.g., 0.5 cm) intervals. All cut surfaces should be examined carefully for tumor nodules [60]. Gross examination of mass lesions in the liver should include the number and size of nodules, observations of macroscopic vascular invasion, the distance of the tumor from inked surgical margins, and an assessment of any macroscopic changes in the non-neoplastic liver, such as cirrhosis and steatosis. Features such as tumor color, consistency, cystic and degenerative changes, and necrosis can also be included in the gross description [61]. The entire tumor should be examined microscopically, especially for tumors up to 2 cm. For larger tumors, at least one block for each 1-cm sample of tumor is recommended. For a diagnosis of cHCC-CCA, both classical HCC and CCA components must be recognized. Therefore, all different tumor areas and transition areas should be sampled [62,63].

### 9.2. Diagnostic Approach

PLC remains a diagnostic challenge, especially for insufficiently sampled tumors. A representative biopsy of the lesion and an adequate amount of well-processed tissue are required for diagnosis. A systemic diagnostic approach for mass lesions facilitates accurate diagnosis [64]. H&E staining is the standard stain for liver pathology [65]. Careful histological evaluation of H&E-stained sections at low-power magnification is the first and most important step for diagnosis. After locating lesional tissue, tumor cell morphology, growth (architectural) pattern, and stromal characteristics should be examined. To aid diagnosis, clinical information, radiologic findings, and AFP and CA19-9 serum levels are helpful. Immunohistochemistry and molecular analyses can further aid in the diagnosis of difficult cases and help identify specific types of PLCs [66].

Diagnosis of cHCC-CCA relies on morphology using H&E and histochemical stains for matrix proteins or mucins [1,2,48]. Additional immunohistochemical stains can be used to establish tumor subtype. Unequivocal histological components of HCC and CCA must be present for a cHCC-CCA diagnosis. Carcinomas positive for both HCC and CCA markers do not qualify as cHCC-CCA [67]. If stem/progenitor cell features/phenotypes are observed, they should be noted in a comment as “stem/progenitor cell features/phenotypes present” (Figure 6A–C). Pathologists should be aware that cHCC-CCAs can encompass a wide morphological and immunophenotypic spectrum of lesions. All kinds of combinations can exist. If combinations of PLC are present, diagnostic terminology should specify which forms of PLC are “combined” (e.g., cHCC-CCA, cHCC-CLC, cCCA-CLC, cHCC-CCA-CLC, cHCC-CCA-ICC, etc.). The pathologic diagnostic algorithm for PLCs is summarized in Figure 7.

### 9.3. Differential Diagnosis

cHCC-CCAs must be distinguished from conventional HCCs and CCAs. Prominent pseudoglands are commonly seen in conventional HCCs and should not be mistaken for true glandular structures in CCAs (leading to a misclassified cHCC-CCA). Proteinaceous material within the pseudoglandular structures may be confused with mucin and mistaken as the CCA component in cHCC-CCA. The presence of bile or proteinaceous material within the pseudoglands helps distinguish them from mucin within true neoplastic glands in CCA [36,68]. Positive immunostains for HepPar-l or arginase-1 are also helpful. Finally, reactive bile ductules within or around HCC may resemble the CCA component in cHCC-CCA. However, reactive bile ductules are commonly accompanied by an inflammatory cell infiltrate and are not mass forming, while the CCA components in cHCC-CCA are usually surrounded by a desmoplastic stroma that lacks inflammatory cells, and are mass forming [69].

Entrapped benign hepatocytes within CCA can be misdiagnosed as the HCC component of cHCC-CCA. Some CCAs showing solid nests and cord-like structures should be differentiated from HCCs [7]. Mucin production and glandular differentiation support the diagnosis of CCA. Immunohistochemistry is required for morphologically equivocal cases. Expression of hepatocytic markers, such as HepPar-1, arginase-1, and glypican-3, suggests HCC rather than CCA. HCCs positive for cholangiocytic markers, such as CK7 and CK19, should not be classified as cHCC-CCA. Some tumors are so poorly differentiated that distinguishing between HCC and CCA may not be possible [67]. ICCs with desmoplastic stroma may be misdiagnosed as CCA. CLC may resemble CCA with ductular configuration. Morphologic diagnosis is key, and careful interpretation of the immunohistochemical stains is needed.

## 10. Future Perspectives

Currently, diagnosis of cHCC-CCA is based on morphological features identified by routine histochemical stains, but diagnosis can be challenging if the tumor is poorly or very poorly differentiated. More practical diagnostic standardization of cHCC-CCA would improve clinical management. Although minimally invasive biomarkers are available for the diagnosis and prognosis of HCC and CCA [70,71,72], none exist for cHCC-CCA. More specific biomarkers and genetic markers that accelerate the detection of cHCC-CCAs are needed.

Surgery is the only curative treatment for cHCC-CCA. The use of systemic therapy and liver transplantation for treating cHCC-CCA remains controversial. A better understanding of the molecular basis of cHCC-CCA will facilitate development of targeted therapeutic agents [73]. Artificial intelligence has emerged as a unique modality with which to improve HCC clinical care, by improving HCC risk prediction, diagnosis, and prognostication [74]. In time, deep learning methods for digital pathology analysis will be an effective way to address multiple clinical questions, from biological understanding and diagnosis to prediction of treatment outcomes [75,76].

## 11. Prognosis

cHCC-CCAs and iCCAs are staged the same in the 8th edition (2017) of the American Joint Committee on Cancer (AJCC) tumor staging system [77]. Resection with lymph node dissection is the common surgical approach. In general, the prognosis for cHCC-CCA is worse than HCC [78,79]. Some studies report that the biological behavior and survival rates of patients with cHCC-CCA are intermediate relative to patients with HCC and CCA [80,81]. cHCC-CCAs more often present with distant spread relative to HCC [82] and have a high risk for recurrence after resection [83]. The main adverse prognostic factors are large tumor size (>5 cm), presence of satellite nodules, lymph node positivity, multiple tumor foci, vascular invasion, high tumor stage, high levels of CA19-9, and surgical margins <2 cm from the tumor [67].

## 12. Conclusions

cHCC-CCA represents a distinctive primary liver malignancy with unequivocal morphologic features of both hepatocytic and cholangiocytic differentiation. The diagnosis of cHCC-CCA should be carefully made based on routine H&E staining. Despite considerable advances in our understanding of cHCC-CCA, its molecular pathogenesis and prognostic and predictive biomarkers remain poorly characterized. Further molecular and genetic characterization studies are needed to better understand the biology of cHCC-CCA and improve disease management.

## Figures and Tables

**Figure 1 biomedicines-10-01826-f001:**
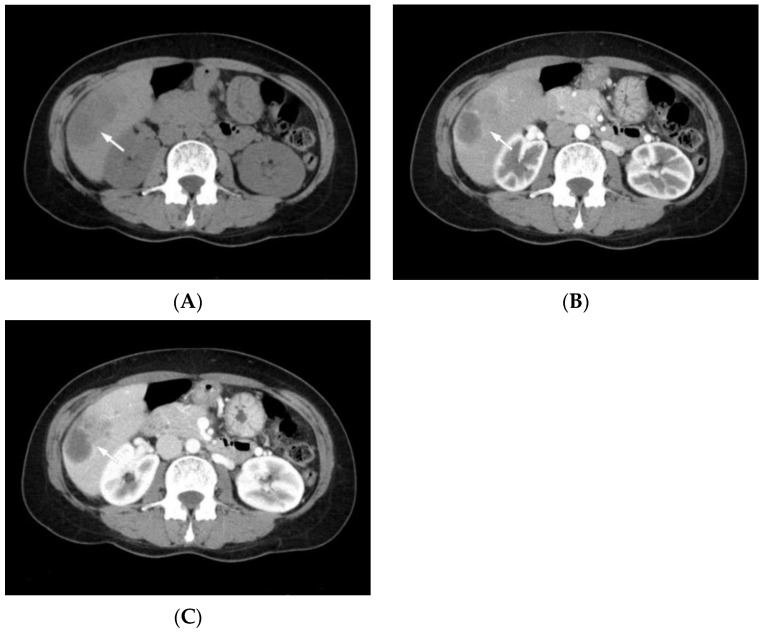
Computed tomography (CT) scan images of combined hepatocellular-cholangiocarcinoma in a 48-year-old female patient with chronic hepatitis B. (**A**) Axial non-contrast CT image shows a hypodense mass (arrow) in the right hepatic lobe. (**B**) Axial contrast-enhanced CT image shows a heterogeneous enhancement (arrow) in the right hepatic lobe during the arterial phase. (**C**) Axial contrast-enhanced CT image shows a heterogeneous enhancement (arrow) in the right hepatic lobe during the portal venous phase.

**Figure 2 biomedicines-10-01826-f002:**
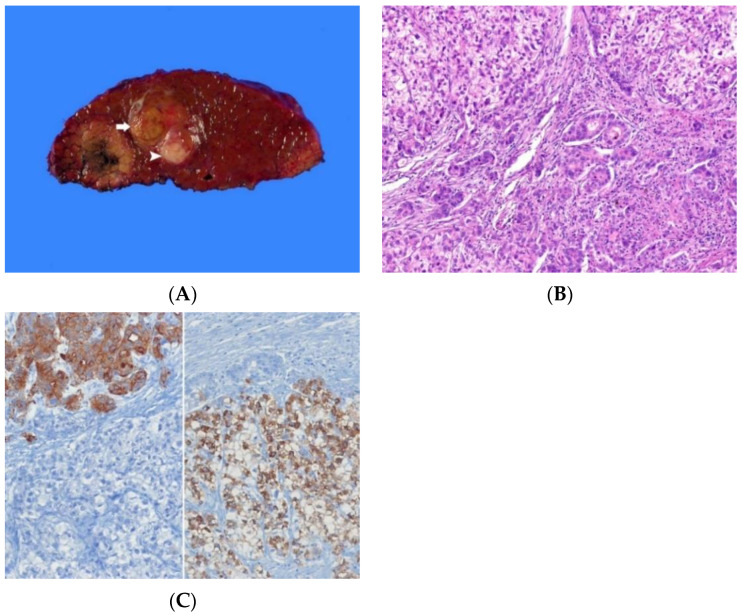
Combined hepatocellular-cholangiocarcinoma. (**A**) In part, the tumor is yellowish-brown and soft (arrow), similar to hepatocellular carcinoma, and is also in part grayish-white and fibrotic (arrowhead), similar to cholangiocarcinoma. The background liver is cirrhotic. (**B**) The tumor shows both a hepatocytic differentiation area of trabecular pattern and a cholangiocytic differentiation area of tubular pattern (H&E stain, ×100). (**C**) The tumor cells show positivity for cytokeratin 19 (left) in cholangiocytic differentiation area and HepPar-1 (right) in hepatocellular differentiation area (immunohistochemical stain for cytokeratin 19 [left] and HepPar-1 [right], ×100).

**Figure 3 biomedicines-10-01826-f003:**
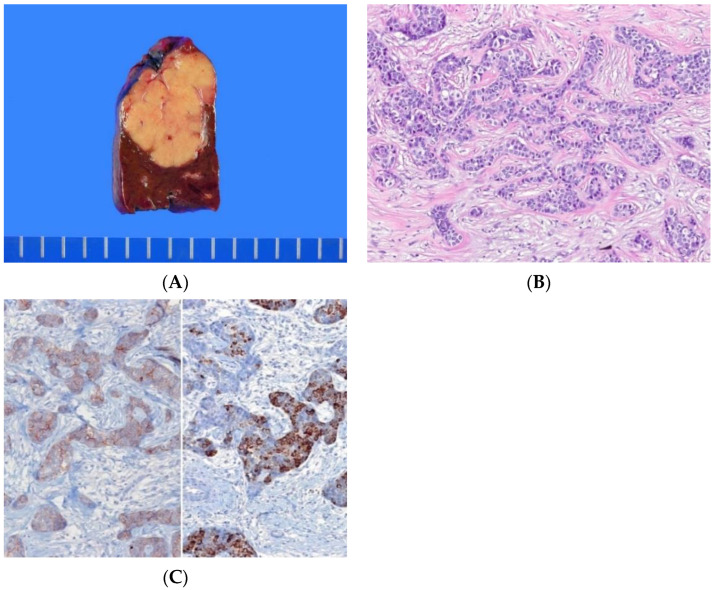
Intermediate cell carcinoma. (**A**) The tumor is subcapsular, well-demarcated, homogeneously grayish-white, and fibrotic. (**B**) The tumor cells show intermediate morphology between hepatocytes and cholangiocytes. They have monomorphic, round nuclei and a small amount of eosinophilic cytoplasm and are arranged in trabecular or cord-like pattern in fibrous stroma (H&E stain, ×100). (**C**) The tumor cells show simultaneous expression for the cholangiocytic marker cytokeratin 19 (left) and the hepatocytic marker HepPar-1 (right) (immunohistochemical stain for cytokeratin 19 [left] and HepPar-1 [right], ×100).

**Figure 4 biomedicines-10-01826-f004:**
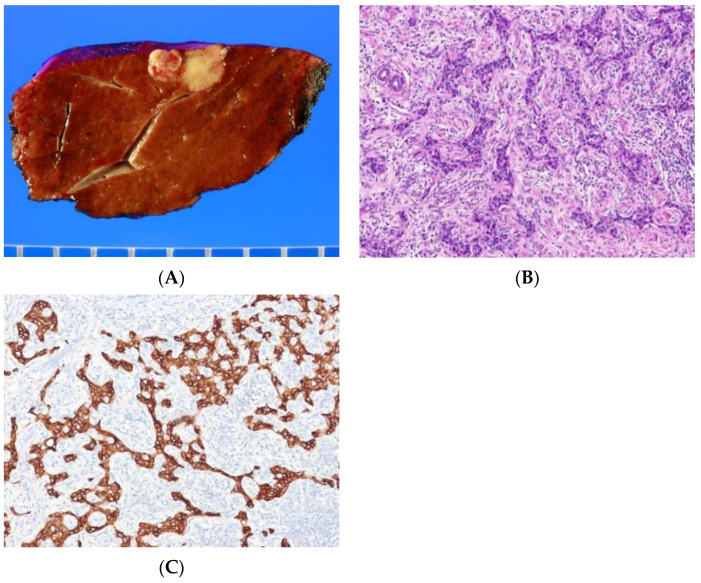
Cholangiolocarcinoma. (**A**) The tumor is light yellowish-white and infiltrative. (**B**) The tumor shows small cuboidal cells resembling the canal of Hering or bile ductule, with anastomosing pattern (antler-like pattern) in fibrous stroma (H&E stain, ×100). (**C**) The tumor cells are positive for cytokeratin 19 (immunohistochemical stain for cytokeratin 19, ×100).

**Figure 5 biomedicines-10-01826-f005:**
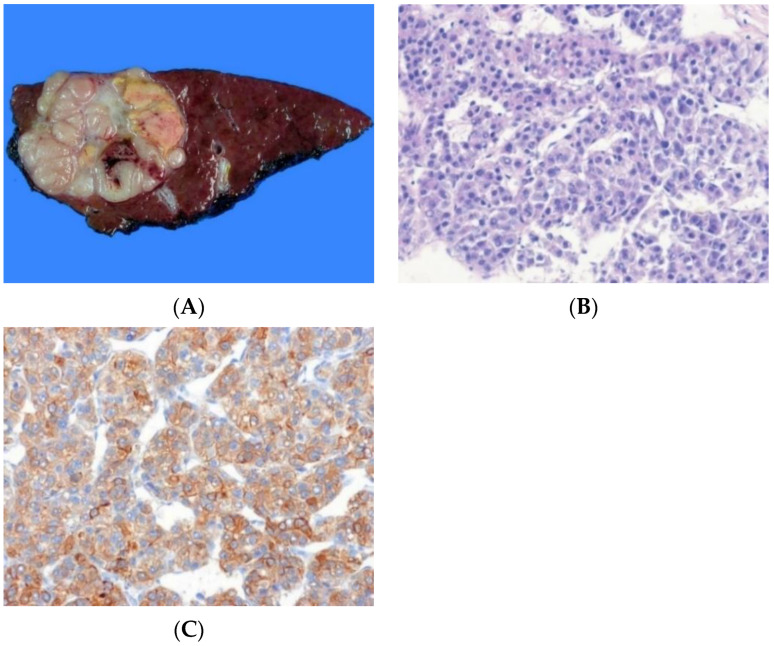
Hepatocellular carcinoma with cytokeratin 19 expression. (**A**) The tumor is well-demarcated, lobulated, and grayish-white, with focal hemorrhage and necrosis. (**B**) The tumor shows typical hepatocellular carcinoma features with polygonal cells arranged in a thickened trabecular pattern (H&E stain, ×100). (**C**) The tumor cells are diffusely positive for cytokeratin 19 (immunohistochemical stain for cytokeratin 19, ×100).

**Figure 6 biomedicines-10-01826-f006:**
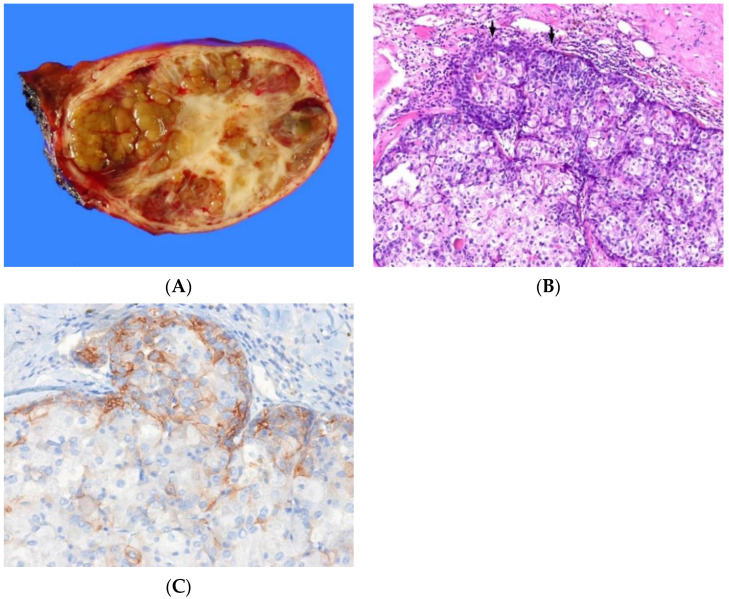
Hepatocellular carcinoma with stem/progenitor cell features/phenotypes. (**A**) The tumor is encapsulated, light green, and soft, with central grayish-white, fibrotic areas. (**B**) The tumor shows solid nests composed of hepatocellular tumor cells. Stem/progenitor cells (arrows) with dark nuclei and scanty cytoplasm are present in the periphery of hepatocellular carcinoma trabeculae. The fibrous stroma surrounds the tumor cell nests (H&E stain, ×100). (**C**) The stem/progenitor cells are positive for the stem/progenitor cell marker cytokeratin 19 (immunohistochemical stain for cytokeratin 19, ×200).

**Figure 7 biomedicines-10-01826-f007:**
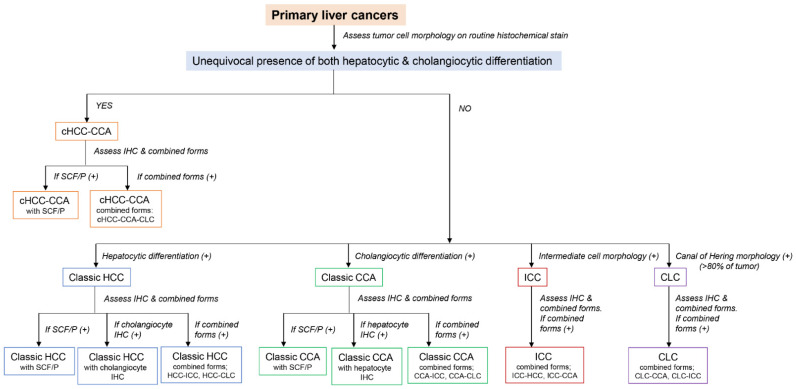
Pathological diagnostic algorithm of primary liver cancers according to tumor cell morphology, immunohistochemistry (IHC), and combined forms. HCC, hepatocellular carcinoma; CCA, cholangiocarcinoma; cHCC-CCA, combined hepatocellular-cholangiocarcinoma; ICC, intermediate cell carcinoma; CLC, cholangiolocarcinoma.

**Table 1 biomedicines-10-01826-t001:** Evolution of the WHO classification of cHCC-CCA.

	2000 WHO Classification (3rd Edition)	2010 WHO Classification (4th Edition)	2019 WHO Classification (5th Edition)
Tumor category	Malignant epithelial tumors	Malignancies of mixed or uncertain origin	Malignant biliary tumors
Tumor entities or subtypes	cHCC-CCA	cHCC-CCA, classical type	cHCC-CCA ^(b)^
		cHCC-CCA with stem cell features ^(a)^, typical subtype	
		cHCC-CCA with stem cell features, intermediate-cell type	Intermediate cell carcinoma ^(c)^
		cHCC-CCA with stem cell features, cholangiolocellular type	Cholangiolocarcinoma ^(d)^

cHCC-CCA, combined hepatocellular-cholangiocarcinoma; ^(a)^ the diagnostic category ‘‘combined hepatocellular-cholangiocarcinoma with stem cell features’’ is no longer recommended; ^(b)^ defined by the unequivocal presence of both hepatocytic and cholangiocytic differentiation within the same tumor. No minimum cut-off amount of each component; ^(c)^ diagnosis should be reserved for primary liver carcinomas in which monotonous intermediate features between hepatocytes and cholangiocytes are present in the entire tumor; ^(d)^ the accepted criterion for cholangiolocarcinoma (CLC) is that >80% of tumors consist of CLC. CLC without an HCC component is now considered a subtype of small duct intrahepatic CCA. CLC can be a component of cHCC-CCA if an HCC component is present.

**Table 2 biomedicines-10-01826-t002:** Histologic features of HCC, CCA, cHCC-CCA, stem/progenitor cell, ICC, and CLC.

	HCC	CCA	cHCC-CCA	Stem/Progenitor Cell	ICC	CLC
Tumor cell morphology	Polygonal tumor cells with round nuclei and abundant eosinophilic cytoplasm	Small to medium-sized, cuboidal or columnar cells with palely eosinophilic or vacuolated cytoplasm	Tumor cell morphology showing both unequivocal hepatocytic and cholangiocytic differentiation	Small uniform cells with hyperchromatic nuclei, scant cytoplasm, and a high nuclear/ cytoplasmic ratio	Tumor cells are smaller than normal hepatocytes, but larger than stem/progenitor cells; monotonous intermediate features between hepatocytes and cholangiocytes	Tumor cells resembling cholangioles (or canals of Hering); usually much smaller than normal hepatocytes and relatively less cytoplasm
Architecture	Trabecular, solid, pseudoglandular pattern	Glandular or tubular pattern with a variable-sized lumen, solid, cord-like, or micropapillary pattern	Two components are either close to each other or intermingled; the transition between them can be poorly defined or sharp	Small nests	Trabeculae, cords, solid nests, or strands	Tubular, cord-like, anastomosing pattern (antler-like pattern) or thin, malignant ductular-like structure
Bile	Present	Absent	Present	Absent	Absent	Absent
Mucin	Absent	Present	Present	Absent	Absent	Absent
Other histologic features	Steatosis, Mallory-Denk bodies, hyaline bodies, pale bodies	Frequently abundant fibrous stroma	Transitional area between HCC and CCA components shows mixed features with intermediate morphology	Most often found at interface between a nest of carcinoma and the adjoining tumoral desmoplastic stroma	Marked desmoplastic or acellular hyalinized stroma	Densely hyalinized stroma; may show trabecular and replacing growth at its interface with the surrounding nontumorous liver

HCC, hepatocellular carcinoma; CCA, cholangiocarcinoma; cHCC-CCA, combined hepatocellular-cholangiocarcinoma; ICC, intermediate cell carcinoma; CLC, cholangiolocarcinoma.

**Table 3 biomedicines-10-01826-t003:** Immunohistochemical markers and albumin mRNA in-situ hybridization for identifying hepatocytic, cholangiocytic differentiation, and stem/progenitor cells.

	Marker	Staining Pattern	Approximate Positivity	Comment
Hepatocytic differentiation	Arginase-1	Nuclear & cytoplasmic	90% of HCC	Better than HepPar-1 in poorly differentiated HCC
	HepPar-1	Cytoplasmic	90% of HCC	Better than Arginase-1 in well differentiated HCC
	Glypican-3	Cytoplasmic	70–90% of HCC	Poorly differentiated HCCs are more likely to be positive
	Polyclonal CEA	Canalicular	60–80% of HCC	Poorly differentiated HCCs are frequently negative
	CD10	Canalicular	60–80% of HCC	Poorly differentiated HCCs are frequently negative
	Alpha-fetoprotein	Cytoplasmic	30% of HCC	Well differentiated HCCs are frequently negative
	Albumin mRNA in situ hybridization	Cytoplasmic	>95% of HCC	
Cholangiocytic differentiation	Cytokeratin 7	Cytoplasmic	90% of CCA	
	Cytokeratin 19	Cytoplasmic	80–90% of CCA	
	EpCAM (MOC31)	Membrane	80–90% of CCA	
	CA19-9	Cytoplasmic	60% of CCA	
	Albumin mRNA in situ hybridization	Cytoplasmic	50–90% of CCA	90% of small duct type; 50% of large duct type
Stem/progenitor cells	CK19	Cytoplasmic		
	EpCAM (MOC31)	Membrane		
	CD56 (NCAM)	Cytoplasm		
	CD117 (KIT)	Cytoplasmic		
	CD133	Cytoplasm		
	SALL4	Nuclear		

HCC, hepatocellular carcinoma; CCA, cholangiocarcinoma; EpCAM, epithelial cell adhesion molecule; CA19-9, carbohydrate antigen 19-9; SALL4, spalt-like transcription factor 4.

**Table 4 biomedicines-10-01826-t004:** Classification and histological relationships of primary liver carcinomas.

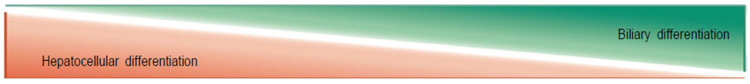
Tumor Classification	Hepatocellular Carcinoma	cHCC-CCA	ICC	CLC	Intrahepatic Cholangiocarcinoma
HCC	HCCwith SPCF/P	HCC with Cholangiocyte IHC Expression	cHCC-CCA	cHCC-CCAwith SPCF/P	iCCAwith Hepatocyte IHC Expression	iCCAwith SPCF/P	iCCA
Histologicalfeatures	Typical HCC	Typical HCC with SPCF	Typical HCC	Typical HCC & CCA	Typical HCC & CCAwith SPCF	Typical intermediate cell features	Typical CLC features (>80% of tumor cells)	Typical CCA	Typical CCA with SPCF	Typical CCA
Immuno-histochemicalfeatures	Hepatocytic markers (+)	Hepatocytic markers (+) & stem/progenitor cell makers (+)	Hepatocytic markers (+) &cholangiocytic markers (+)	Hepatocytic markers (+) & cholangiocytic markers (+)	Hepatocytic markers (+), cholangiocytic markers (+) & stem/progenitor cell markers (+)	Hepatocytic markers (+) & cholangiocytic markers (+)	Cholangiocytic markers (+) & CD56 (NCAM) (+), CD117 (KIT) (+)	Cholangiocytic markers (+) & hepatocytic markers (+)	Cholangiocytic markers (+) & stem/progenitor cell makers (+)	Cholangiocytic markers (+)

Primary liver carcinomas (PLCs) show a broad spectrum of histological and immunohistochemical features from hepatocellular (left) through combined (middle) to biliary differentiation (left). The classification of PLCs depends on the relative proportions of hepatocellular and biliary differentiation features, stem/progenitor cell features/phenotypes, and IHC expression. HCC, hepatocellular carcinoma; iCCA, intrahepatic cholangiocarcinoma; cHCC-CCA, combined hepatocellular-cholangiocarcinoma; ICC, intermediate cell carcinoma; CLC, cholangiolocarcinoma; IHC, immunohistochemistry; SPCF/P, stem/progenitor cell features or phenotypes.

**Table 5 biomedicines-10-01826-t005:** Common recurrent mutations in hepatocellular carcinoma, intrahepatic cholangiocarcinoma, and cHCC-CCA.

Hepatocellular Carcinoma	Intrahepatic Cholangiocarcinoma	cHCC-CCA
*TP53* mutations (60%) *TERT* gene promoter mutations (50–60%) *CTNNB1* mutations (40%)	*KRAS* mutations (20% of large duct type; ~0% of small duct type) *TP53* mutations (30% of large duct type) *IDH1* mutations (15% of small duct type; ~0% of large duct type) *FGRF2* translocation (10% of small duct type; ~0% of large duct type) *ARID1A* mutations *BAP1* mutations *PBRM1* mutations	*TP53* mutations (80%) *TERT* promoter mutations (80%) *KRAS* mutations (55%) *CTNNB1* mutations (20%) *AXIN1* mutations (20%) *IDH1* mutations *KMT2D* mutations

cHCC-CCA, combined hepatocellular-cholangiocarcinoma; *TERT*, telomerase reverse transcriptase; *CTNNB1,* catenin beta 1; *IDH1,* isocitrate dehydrogenase 1, *FGFR2*, fibroblast growth factor receptor 2; *ARID1A*, AT-rich interaction domain 1A; *BAP1,* BRCA1 associated protein 1; *PBRM1*, polybromo 1; *KMT2D*: lysine methyltransferase 2D.

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
