# Peer review of "Combined Hepatocellular-Cholangiocarcinoma: An Update on Pathology and Diagnostic Approach"

_biomedicines, 2022, doi:10.3390/biomedicines10081826_

Round 1
Reviewer 1 Report
The authors have appropriately addressed the comments with images and histology.
Reviewer 2 Report
At first glance, the authors have certainly improved the manuscript with the illustrations or additions to the figures. However, I cannot understand the use of an illustration with imaging from someone else's paper. Even if this was done under the Creative Commons License, which would need to be clarified, I find this approach somewhat unfortunate. If the authors want to stand out from the reviews already published on the topic recently, they should present imaging on their own patients to the readership. Surely there should be preoperative imaging on the tumors in Figure 2-5 that the authors could use
Round 2
Reviewer 2 Report
The authors provided now CT scans
This manuscript is a resubmission of an earlier submission. The following is a list of the peer review reports and author responses from that submission.
Round 1
Reviewer 1 Report
Choi and Ro have crafted a very thorough and well-written summary of combined hepatocellular-cholangiocarcinoma in this review. The manuscript is well organized and covers the major updates in this diagnosis.
The major challenge with this manuscript is that I don't see what is new compared with the other reviews that have been published in the last two years. These reviews are included in the references: Momuta et al, and Beaufrere et al to name two of the references.
Reviewer 2 Report
The review by Choi and Ro provides a very nice and stringent overview of combined cHCC-CCAs.
I have only a few comments or suggestions for improvement:
If the authors mention diagnostic imaging in the manuscript, exemplary illustrations of CT or MRI examinations would also be very important and helpful to the readership. Apart from that, I would list the remarks on imaging diagnostics at the beginning of the paper (e.g., after the clinical presentation). In particular, contrast-enhanced ultrasound is also frequently performed in the diagnosis of liver tumors. Are there data on how cHCC-CCAs present in this modality of diagnostic and can be differentiated from HCC and intrahepatic CCC? This information would be quite interesting to the readership.
In the abstract (line 15), the "a" before "diagnostic chalenging" should be deleted
Table 4: "Immunohistochemical features" should be in bold
Please pay attention to the page numbers. These start from 1-5 and then start again at 1. There seems to be a formatting problem here due to the horizontal formatting of the tables